# Characteristics and outcomes of pregnant women with placenta accreta spectrum in Italy: A prospective population-based cohort study

**Sara Ornaghi**[1]*, **Alice Maraschini**[2], **Serena Donati**[2], **on behalf of The Regional Obstetric Surveillance System Working Group**[¶]

1 Department of Obstetrics and Gynecology, Foundation MBBM at San Gerardo Hospital, University of Milan-Bicocca School of Medicine and Surgery, Monza, Italy, 2 National Centre for Disease Prevention and Health Promotion, Italian National Institute of Health, Rome, Italy

¶ The complete membership of The Regional Obstetric Surveillance System Working Group is provided in the Acknowledgments.

* sara.ornaghi@unimib.it

**Data Availability Statement:** Data cannot be shared publicly because of ethical standards and legal requirements. Data are available from the

## Abstract

### Introduction

Placenta accreta spectrum (PAS) is a rare but potentially life-threatening event due to massive hemorrhage. Placenta previa and previous cesarean section are major risk factors for PAS. Italy holds one of the highest rates of primary and repeated cesarean section in Europe; nonetheless, there is a paucity of high-quality Italian data on PAS. The aim of this paper was to estimate the prevalence of PAS in Italy and to evaluate its associated factors, ante- and intra-partum management, and perinatal outcomes. Also, since severe morbidity and mortality in Italy show a North-South gradient, we assessed and compared perinatal outcomes of women with PAS according to the geographical area of delivery.

### Material and methods

This was a prospective population-based study using the Italian Obstetric Surveillance System (ItOSS) and including all women aged 15–50 years with a diagnosis of PAS between September 2014 and August 2016. Six Italian regions were involved in the study project, covering 49% of the national births. Cases were prospectively reported by a trained clinician for each participating maternity unit by electronic data collection forms. The background population comprised all women who delivered in the participating regions during the study period.

### Results

A cohort of 384 women with PAS was identified from a source population of 458 995 maternities for a prevalence of 0.84/1000 (95% CI, 0.75–0.92). Antenatal suspicion was present in 50% of patients, who showed reduced rates of blood transfusion compared to unsuspected patients (65.6% versus 79.7%, $P = 0.003$). Analyses by geographical area showed

Ethics Committee of the Istituto Superiore di Sanità - Italian National Institute of Health (contact via email: segreteria.comitatoetico@iss.it) for researchers who meet the criteria for access to confidential data.

**Funding:** S.D. Italian Ministry of Health. The funders had no role in study design, data collection and analysis, decision to publish, or preparation of the manuscript.

**Competing interests:** The authors have declared that no competing interests exist.

**Abbreviations:** CI, confidence interval; CS, cesarean section; ItOSS, Italian Obstetric Surveillance System; NOSS, Nordic Obstetric Surveillance System; PAS, placenta accreta spectrum; RR, relative risk; UKOSS, United Kingdom Obstetric Surveillance System.

higher rates of both concomitant placenta previa and prior CS (62.1% vs 28.7%, *P*<0.0001) and antenatal suspicion (61.7% vs 28.7%, *P*<0.0001) in women in Southern compared to Northern Italy. Also, these women had lower rates of hemorrhage ≥2000 mL (29.6% vs 51.2%, *P*<0.0001), blood transfusion (64.5% vs 87.5%, *P* = 0.001), and severe maternal morbidity (5.0% vs 11.1%, *P* = 0.036). Delivery in a referral center for PAS occurred in 71.9% of these patients.

## Conclusions

Antenatal suspicion of PAS is associated with improved maternal outcomes, also among high-risk women with both placenta previa and prior CS, likely because of their referral to specialized centers for PAS management.

## Introduction

Placenta accreta spectrum (PAS) is an obstetric condition caused by excessive trophoblast invasion into the myometrium of the uterine wall [1]. Defective decidualization in an area of scarring, mostly due to previous uterine surgery, is supposed to be the main underlying mechanism of PAS [2].

Prevalence of PAS ranges from 0.01 to 1.1% [3, 4], and it has progressively increased due to the raising rate of cesarean sections (CS), and alongside that of placenta previa [4–7]. Placenta previa after prior CS is the most important risk factor for PAS, with 11%, 40%, and 61% rate of PAS in case of placenta previa associated to one, two, or three previous CS, respectively [8–11]. Maternal age ≥35 years, high parity, prior uterine surgeries other than CS, history of infertility, and infertility-related procedures are additional risk factors [12–16].

Although rare, PAS represents a potentially life-threatening event, especially if not suspected before delivery [17, 18]. It may result in massive hemorrhage ultimately requiring emergency hysterectomy to prevent maternal death [19–21]. Thus, PAS can be considered a "near-miss" event [17, 22]. "Near-miss" events are proxies of maternal health care quality, and their monitoring and in-depth investigation provide an essential feedback to improve obstetric care [23].

Considering that hemorrhage is the leading cause of maternal mortality and morbidity in Italy [24, 25], where there is a paucity of high-quality studies on PAS notwithstanding high rates of CS [26–31], the Italian Obstetric Surveillance System (ItOSS) carried out a prospective, population-based study on hemorrhagic "near-miss" events, including PAS.

The aim of this paper is to estimate the incidence of PAS and to analyze its associated factors, management, and perinatal complications. In addition, since Italy has regional health care imbalances with the South displaying higher rates of morbidity and mortality [20, 25], outcomes were compared according to the geographical area of delivery.

## Material and methods

This is a prospective, population-based study including all women aged 15–50 years and delivering at ≥22 weeks of gestation with a diagnosis of PAS from September 2014 to August 2016 in six Italian regions covering 49% of the national births. These regions were selected by annual number of births (≥25 000) and to ensure the representativeness of the Northern (Piedmont, Emilia Romagna, and Tuscany) and Southern (Lazio, Campania and Sicily) areas.

The present study is part of a wider research project on severe maternal morbidity due to obstetric hemorrhage coordinated by the ItOSS, as previously reported [20]. Briefly, the ItOSS project prospectively collected data on women delivering at ≥22 weeks of gestation with any of the following complications: (1) severe postpartum hemorrhage, defined as "hemorrhage within 7 days from delivery requiring ≥4 units of whole blood or packed red blood cells"; (2) "hemorrhage due to complete or incomplete uterine rupture"; (3) "peripartum hysterectomy within 7 days from delivery"; and (4) PAS, clinically defined as "difficult or incomplete manual removal of the placenta following vaginal delivery and the need of blood transfusion within 48 hours" or "difficult removal of the placenta during cesarean delivery and clinical evidence of an abnormally invasive placenta".

The present study includes all cases of PAS as defined in (4), independent of the associated outcomes, such as severe postpartum hemorrhage (1), uterine rupture (2), and peripartum hysterectomy within 7 days from delivery (3), leading to inclusion in the wider ItOSS research project.

All maternity units in the selected regions were invited to participate in the study and to appoint a clinician as reference person for reporting incident cases. Unified electronic data collection forms, prepared by a team of national experts by adapting the forms of the Nordic Obstetric Surveillance Study [32], were used for data collection. Each reference person was trained to use the web system for data collection before study's commencement, and received a monthly reminder by email to promote complete reporting. A multidisciplinary audit involving all healthcare professionals that assisted the women with PAS diagnosis was recommended in each participating maternity unit.

## Statistical analyses

The prevalence rate was calculated as the number of PAS per 1000 maternities with a 95% CI, assuming the Poisson approximation to the binomial distribution. When available, the background population was retrieved from the National Hospital Discharge database by selecting all women aged 15–50 who delivered during the same study period in the participating maternity units of the selected regions. When not available, the background population was estimated in aggregate form from the National Birth Register, year 2015 [33].

Potential factors associated to PAS were identified by calculating unadjusted relative risks (RR) and 95% CI. Dichotomous data were compared using Pearson Chi-square test or Fisher exact test for categorical variables and Mann-Whitney test for continuous variables.

Analyses were performed using SPSS 26.0 (IBM Corp., NY, USA) and Stata/MP 14.2 (Stata Corp., TX, USA).

## Ethical approval

The study was approved by the Ethics Committee of the Italian National Institute of Health (Prot. PRE-839/13). Data were fully anonymized before being accessed and analyzed. Thus, need for informed consent was waived by the local Ethics Committee.

## Results

Seven of the 212 maternity units in the six selected regions did not provide the requested data, for a 96.7% participation rate.

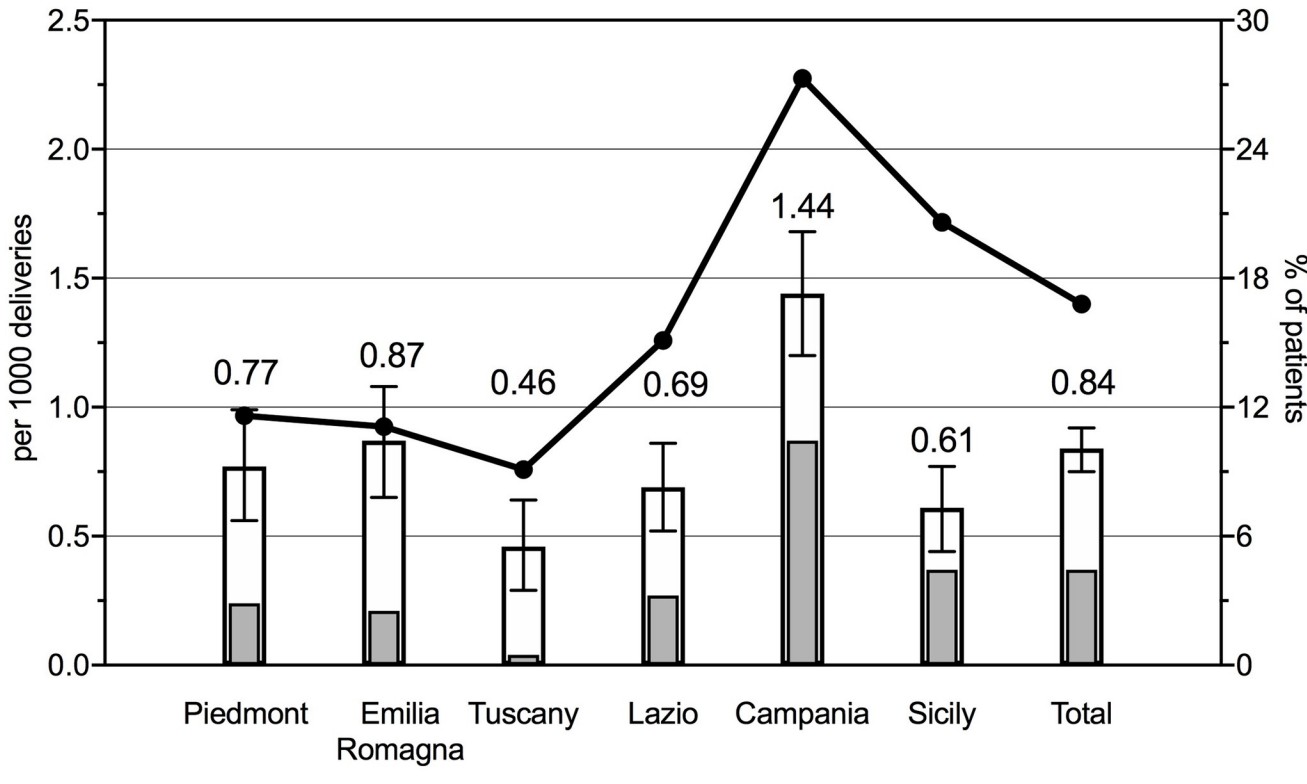

**Fig 1. Regional and overall PAS rate and percentage of previous cesarean section in the background population.** Bar graphs show regional and overall prevalence distribution of PAS with the 95% CI (white bars and black lines). Grey bars display the contribution to the regional and overall PAS rate given by women with both placenta previa or low-lying and previous cesarean section. Both white and grey bars are plotted on the left Y axis. Solid line with dots shows regional and overall rate of previous cesarean section in the background population (plotted on the right Y axis).

### PAS rate and associated factors

During the study period, there were 372 cases of PAS notified. Assessment of data completeness led to recovery of twelve additional cases, for a total of 384 cases out of 458 995 maternities with an estimated prevalence of 0.84 per 1000 (95% CI, 0.75–0.92).

Along with the regional and overall estimates of the PAS rate, Fig 1 shows the contribution to the PAS rate given by women with both placenta previa or low-lying and previous CS. The solid line describes the percentage of women with previous CS in the background population.

Women with PAS had a median age of 35 years (IQR, 31.4–39.0 years) at delivery; six women were older than 45 and one younger than 20 years (Table 1). PAS patients were mostly Italian with a low education level and more likely multiparas. Overall, 54% had a previous CS, with 18.5% and 6.5% having two or ≥ three previous CS, respectively. Placenta previa or low-lying was diagnosed in 60% of women. In 44.6% of cases there was both a placenta previa or low-lying and a prior CS.

**Table 1. Distribution of PAS by maternal sociodemographic characteristics, obstetric history, and current pregnancy course and outcomes.**

| Variables | | | PAS N = 384 | | Background population [a] N = 458 995 | | Rate ‰ 0.84 | RR (95% CI) |
|---|---|---|---|---|---|---|---|---|
| | | | N | % | N | % | | |
| **Maternal characteristics** | Maternal age | • <35 years | 184 | 47.9 | 309 940 | 67.5 | 0.59 | ref. |
| | | • ≥35 years | 200 | 52.1 | 149 055 | 32.5 | 1.34 | 2.26 (1.85–2.76) |
| | Citizenship (5 missing) | • Italian | 308 | 80.2 | 373 258 | 81.3 | 0.83 | ref. |
| | | • Not Italian | 71 | 18.5 | 85 737 | 18.7 | 0.83 | 1.00 (0.78–1.30) |
| | Education level [b] (44 missing) | • High | 75 | 19.5 | 126 683 [c] | 27.6 | 0.59 | ref. |
| | | • Low | 265 | 69.0 | 332 313 [c] | 72.4 | 0.80 | 1.35 (1.04–1.74) |
| | Smoking during pregnancy (119 missing) | • No | 227 | 59.1 | NA | NA | | |
| | | • Yes | 38 | 9.9 | | | | |
| | Pregestational BMI (51 missing) | • <30 Kg/m$^2$ | 301 | 78.4 | NA | NA | | |
| | | • ≥30 Kg/m$^2$ | 32 | 8.3 | | | | |
| **Obstetric history** | Parity (5 missing) | • Nulliparas | 122 | 31.8 | 251 988 | 54.9 | 0.48 | ref. |
| | | Multiparas | 257 | 66.9 | 207 007 | 45.1 | 1.24 | 2.56 (2.07–3.18) |
| | | • ≥3 | 37 | 9.6 | 6 960 [c] | 1.52 | 5.32 | 4.81 (3.40–6.81) |
| | Previous CS and/or uterine surgery [d] (5 missing) | • Neither one | 109 | 28.4 | 333 919 | 72.8 | 0.33 | ref. |
| | | • Uterine surgery, no CS | 63 | 16.4 | 47 827 | 10.4 | 1.32 | 4.04 (2.96–5.50) |
| | | • CS, no uterine surgery | 133 | 34.6 | 61 414 | 13.4 | 2.17 | 6.63 (5.15–8.55) |
| | | • 1 | 63 | 16.4 | 50 627 | 11.0 | 1.24 | 3.81 (2.80–5.20) |
| | | • ≥2 | 62 | 16.1 | 10 786 | 2.4 | 5.75 | 17.61 (12.90–24.04) |
| | Previous PPH | • No | 369 | 96.1 | NA | NA | | |
| | | • Yes | 15 | 3.9 | | | | |
| **Current pregnancy** | ART (9 missing) | • No | 347 | 90.4 | 450 825 [c] | 98.2 | 0.77 | ref. |
| | | • Yes | 28 | 7.3 | 8 170 [c] | 1.8 | 3.43 | 4.45 (3.03–6.54) |
| | Multiple gestation (18 missing) | • No | 354 | 92.2 | 451 766 | 98.4 | 0.78 | ref. |
| | | • Yes | 12 | 3.1 | 7229 | 1.6 | 1.66 | 2.12 (1.19–3.76) |
| | Placenta previa or low-lying | • No | 156 | 40.6 | 456 067 | 99.4 | 0.34 | ref. |
| | | • Yes | 228 | 59.4 | 2 928 | 0.6 | 77.9 | 227.65 (186.31–278.16) |
| **Delivery** | Geographical location of delivery | • Northern Italy | 136 | 35.4 | 190 018 | 41.4 | 0.72 | ref. |
| | | • Southern Italy | 248 | 64.6 | 268 977 | 58.6 | 0.92 | 1.29 (1.05–1.59) |
| | Mode of delivery (7 missing) | • Vaginal delivery | 91 | 23.7 | 282 232 [c] | 61.5 | 0.32 | ref. |
| | | • spontaneous | 74 | 19.3 | 266 893 [c] | 58.1 | 0.28 | 0.86 (0.63–1.17) |
| | | • operative | 17 | 4.4 | 15 339 [c] | 3.34 | 1.11 | 3.44 (2.05–5.77) |
| | | • Cesarean section | 286 | 74.5 | 176 763 [c] | 38.5 | 1.62 | 5.02 (3.96–6.35) |
| | | • emergent/urgent | 76 | 19.8 | 63 548 [c] | 13.8 | 1.20 | 3.71 (2.74–5.03) |
| | | • elective | 210 | 54.7 | 113 215 [c] | 24.7 | 1.85 | 5.75 (4.50–7.36) |
| | Gestational age at delivery (12 missing) | • ≥37 wks | 175 | 45.6 | 427 490 [c] | 93.1 | 0.41 | ref. |
| | | • <37 wks | 197 | 51.3 | 31 505 [c] | 6.9 | 6.25 | 15.27 (12.47–18.72) |
| | | • 32–36 wks | 172 | 44.8 | 27 062 [c] | 5.9 | 6.36 | 15.53 (12.58–19.16) |
| | | • 22–31 wks | 25 | 6.5 | 4 443 [c] | 1.0 | 5.63 | 13.57 (9.05–20.88) |

BMI, Body Mass Index; CS, cesarean section; PPH, postpartum hemorrhage; ART, Assisted Reproductive Technology.

[a] Source: National Discharge Register;

[b] Education level: high ≥ university degree, low < university degree;

[c] Source: National Birth Register year 2015 for the six Italian regions involved in the study;

[d] Uterine surgery included dilation & curettage, surgical termination of pregnancy, endometrial ablation, operative hysteroscopy, myomectomy, and metroplasty.

Among the 122 (31.8%) nulliparous patients, 42.6% were ≥35 years old, 46.7% had either a history of uterine surgery or an ART-conceived pregnancy, and 28.7% had a placenta previa or low-lying. In 35/122 women, none of these factors was identified.

Delivery occurred in facilities in Southern Italy in 65% of the cases. These women showed higher rates of Italian citizenship (89.3 vs 66.9%, $P<0.0001$), low education level (81.0 vs 72.6%, $P = 0.048$), multiparity (75.3 vs 54.4%, $P<0.0001$), previous CS (65.8% vs 34.6%, $P<0.0001$), placenta previa or low-lying (72.6% vs 35.3%, $P<0.0001$), and a combination of the last two conditions (62.1% vs 28.7%, $P<0.0001$). In turn, PAS pregnancies in Northern Italy were more commonly conceived by ART (11.9% vs 5.0%, $P = 0.023$).

There were 74.5% deliveries by CS, with elective surgery being the most common (73.4%). Median gestational age at delivery was 36 weeks (IQR, 35–38 weeks). Preterm delivery <37 weeks' gestation occurred in 51.3% of cases, and was more frequent among women with placenta previa or low-lying compared to women without (66.0% vs 16.3%, $P<0.0001$).

The analysis of maternal characteristics showed a substantially higher risk of PAS in women with placenta previa or low-lying and with previous CS or other uterine surgery, with the greatest risk increase for ≥2 previous CS (RR 17.6; 95% CI, 12.9–24.0). A modest risk increase was also observed for maternal age ≥35 years, multiparity, low education level, and delivery in Southern Italy (Table 1). Also, ART and multiple gestation significantly increased PAS risk. In addition, women with PAS showed a 5- and 15-fold increase in the odds of delivering by CS and <37 weeks' gestation, respectively.

## Pregnancy, delivery, and perinatal outcomes of PAS

PAS was antenatally suspected in 50% of the cases, more likely in multiparas with prior uterine surgery, placenta previa or low-lying, or a combination of both (Table 2). These conditions were more frequent among women in Southern Italy, and, accordingly, a higher rate of antenatal suspicion was identified (61.7% vs 28.7%, $P<0.0001$).

Most of the suspected women delivered in a high-level hospital setting and by a scheduled CS. None of the 35 women without risk factors for PAS was suspected prenatally, and 26 (74.3%) of them delivered vaginally.

Overall, 32% of patients experienced severe postpartum hemorrhage (PPH) ≥2000 mL. Although suspicion did not impact blood loss (median, IQR: 1500 mL, 1000–2000 mL vs 1500 mL, 1000–2100 mL; $P = 0.226$), it influenced PPH management, with higher rates of surgical treatment, including hysterectomy, in suspected cases (Table 2). Overall, 49.7% of women underwent hysterectomy, mostly as peripartum procedure (95.3%). Damage of adjacent organs during hysterectomy occurred in 23/191 women, more frequently when PAS was suspected (16.9% vs 5.2%, $P = 0.032$). Similarly, patients with antenatal suspicion were more likely to experience post-surgery complications (n = 24, 18.9% vs n = 5, 7.9%; $P = 0.035$), including urological (n = 27) and vascular (n = 2). Of note, all 23 women with complications during surgery and 28/29 women with post-hysterectomy complications had a previous CS in their obstetric history. Also, in 21/23 and 26/29 patients there was a placenta previa or low-lying.

Four women were managed conservatively: three had a partial placenta accreta diagnosed after delivery and only the abnormally adherent cotyledon was left *in situ* whereas the remaining one had an antenatal diagnosis of complete placenta previa with signs of percretion and no attempt of placenta removal was performed at the time of CS. Follow up of these patients was not available at the time of data collection.

Almost 73% of women were transfused with RBC units, with higher rates among unsuspected women (Table 2).

**Table 2. Maternal characteristics, PAS management, and perinatal outcomes stratified according to antenatal suspicion of PAS.**

| Variables | | Antenatal suspicion of PAS | | | | | | P-value |
|---|---|---|---|---|---|---|---|---|
| | | Total | | Yes | | No | | |
| | | N = 384 | | N = 192 (50%) | | N = 192 (50%) | | |
| | | N | % | N | % | N | % | |
| **Obstetric data** | Multiparity (5 missing) | 257 | 66.9 | 166 | 86.5 | 91 | 47.4 | <0.0001 |
| | Previous CS and/or uterine surgery (5 missing) | 270 | 70.3 | 165 | 85.9 | 105 | 54.7 | <0.0001 |
| | Placenta previa or low-lying | 228 | 59.4 | 186 | 96.9 | 42 | 21.9 | <0.0001 |
| | Previous CS and placenta previa or low-lying (5 missing) | 169 | 44.0 | 144 | 75.0 | 25 | 13.0 | <0.0001 |
| **Delivery location, mode, and timing** | Hospital with ≥1 000 annual births | 260 | 67.9 | 127 | 66.1 | 133 | 69.6 | 0.512 |
| | Hospital with <500 annual births | 21 | 5.5 | 4 | 2.1 | 17 | 8.9 | 0.003 |
| | High-level hospital setting (20 missing)[a] | 190 | 49.5 | 128 | 66.7 | 62 | 32.3 | <0.0001 |
| | Mode of delivery (7 missing) | | | | | | | <0.0001 |
| | • Vaginal spontaneous | 74 | 19.3 | 0 | 0 | 74 | 40.0 | |
| | • Vaginal operative | 17 | 4.4 | 0 | 0 | 17 | 9.2 | |
| | • Urgent/emergent CS | 76 | 19.8 | 36 | 18.8 | 40 | 21.6 | |
| | • Elective CS | 210 | 54.7 | 156 | 81.3 | 54 | 29.2 | |
| | Preterm delivery <37 weeks (12 missing) | 197 | 51.3 | 141 | 74.6 | 56 | 30.6 | <0.0001 |
| | • 32–36 weeks | 172 | 44.8 | 127 | 67.2 | 45 | 24.6 | |
| | • 22–31 weeks | 25 | 6.5 | 14 | 7.4 | 11 | 6.0 | |
| **Entity and management of PAS-related hemorrhage** | Blood loss ≥2000 mL (61 missing) | 123 | 32.0 | 62 | 32.3 | 61 | 31.8 | 0.513 |
| | *First- and second-line uterotonic drugs* | | | | | | | |
| | Oxytocin | 188 | 49.0 | 87 | 45.3 | 101 | 52.6 | 0.184 |
| | Prostaglandins | 68 | 17.7 | 23 | 12 | 45 | 23.4 | 0.005 |
| | Methylergometrine | 8 | 2.1 | 0 | 0 | 8 | 4.2 | 0.007 |
| | *Mechanical and surgical procedures* | | | | | | | |
| | Manual removal of the placenta and/or uterine curettage | 48 | 12.5 | 4 | 2.1 | 32 | 16.7 | <0.0001 |
| | Uterine tamponade | 121 | 31.5 | 54 | 28.1 | 67 | 34.9 | 0.187 |
| | Uterine artery embolization | 100 | 26.0 | 87 | 45.3 | 13 | 6.8 | <0.0001 |
| | Intravascular tamponade | 4 | 1.0 | 4 | 2.1 | 0 | 0 | 0.123 |
| | Uterine hemostatic sutures | 44 | 11.5 | 28 | 14.6 | 16 | 8.3 | 0.077 |
| | Vascular hemostatic sutures | 6 | 1.6 | 4 | 2.1 | 2 | 1.0 | 0.685 |
| | Hysterectomy | 191 | 49.7 | 127 | 66.1 | 64 | 33.3 | <0.0001 |
| | *Blood products* | | | | | | | |
| | RBC unit transfusion | 279 | 72.7 | 126 | 65.6 | 153 | 79.7 | 0.003 |
| | • ≥4 RBC units | 102 | 26.6 | 59 | 30.7 | 43 | 22.4 | 0.083 |
| | Plasma transfusions | 101 | 26.3 | 44 | 22.9 | 57 | 29.7 | 0.164 |
| | Platelets transfusion | 14 | 3.6 | 6 | 3.1 | 8 | 4.2 | 0.787 |
| | Fibrinogen | 23 | 6.0 | 11 | 5.7 | 12 | 6.3 | 1.000 |
| **Perinatal outcomes** | ICU admission (8 missing) | 92 | 24.5 | 43 | 22.4 | 49 | 26.6 | 0.401 |
| | Severe maternal morbidity (8 missing)[b] | 27 | 7.2 | 11 | 5.7 | 16 | 8.7 | 0.319 |
| | Maternal mortality (10 missing) | 1 | 0.3 | 0 | 0 | 1 | 0.5 | 1.000 |
| | Perinatal mortality (13 missing) [c] | 8 | 2.1 | 3 | 1.6 | 5 | 2.6 | 0.592 |

CS, cesarean section; RBC, red blood cell; ICU, intensive care unit.

[a] High-level hospital setting was defined as a hospital with availability of ICU and interventional radiology, and possibility of blood transfusion within 15 minutes;

[b] Severe maternal morbidity included vegetative state (n = 1), cardiac arrest (n = 2), respiratory distress (n = 3), acute pulmonary edema (n = 2), disseminated intravascular coagulopathy (n = 6), acute renal failure (n = 1), deep vein thrombophlebitis or pulmonary embolism (n = 1), sepsis or septic shock (n = 1), hemorrhagic shock (n = 7). Damage to adjacent organs during surgery and post-operative complications are described separately (details in the text);

[c] Lazio region excluded.

At least one severe maternal morbidity condition was identified in 27 (7.0%) women, with hemorrhagic shock (n = 7) and disseminated intravascular coagulopathy (DIC, n = 6) being the most frequent. Twelve (3.1%) patients were assisted with mechanical ventilation, and 24% required admission to the ICU. There were no differences in rate of severe maternal morbidity or ICU admission between women with and without suspected PAS (Table 2). Overall, there were 51 (13.3%) patients experiencing organ damage during surgery, post-surgical complications, or a severe maternal morbidity condition.

There was one maternal death in the study cohort, for a fatality rate of 2.6‰; it occurred in a primiparous young woman with no risk factors for PAS and no antenatal diagnosis, who delivered vaginally and experienced uterine inversion in the attempt of removing a partially attached placenta with subsequent severe PPH, DIC, cardiac arrest, and death.

Among 398 infants who were given birth to (372 singletons, ten twins, and two triplets), eight perinatal deaths were identified, for a perinatal mortality rate of 20.1‰: seven (87.5%) cases happened postnatally and in 85.7% of them delivery was before 26 weeks' gestation.

Assessment of PAS management and perinatal outcomes according to geographical area showed that women in Southern Italy were less likely to bleed ≥2000 mL (29.6% vs 51.2%, $P<0.0001$), receive RBC units (64.5% vs 87.5%, $P = 0.001$), be admitted to the ICU (16.5% vs 38.8%, $P<0.0001$), and experience severe maternal morbidity (5.0% vs 11.1%, $P = 0.036$). In turn, hysterectomy was more frequently performed (59.3% vs 32.4%, $P<0.0001$), although with lower rates of intra- and post-operative complications (9.5% vs 25.6%, $P = 0.014$ and 11.6% vs 27.9%, $P = 0.015$). Analysis of delivery location among women with suspected PAS in Southern Italy showed that 71.9% of them gave birth in a referral center.

### Histology data

Histology report was available at the time of data retrieval in 179 cases, 77.1% of whom had undergone hysterectomy.

Overall, PAS was confirmed in 130 (72.6%) patients; depth of invasion with rates of placenta previa or low-lying and previous CS are shown in Fig 2. All histological diagnoses were performed on both uterine and placental specimen, except for five (3.8%) cases with placenta accreta which were identified by assessment of just the placenta.

PAS was antenatally suspected in 62.7%, 52.4%, and 79.4% of cases with placenta accreta, increta, and percreta, respectively. Women with antenatal diagnosis did not show higher grade of invasion, such as placenta increta or percreta, compared to unsuspected women (44.7% vs 37.8%, p = 0.463). However, when analysis was performed by geographical area, women in Southern regions were more likely to have more severe forms of PAS than women in the North (46.5% vs 29.0%, $P = 0.045$).

Rates of PPH ≥2000 mL and of blood transfusion among women with either placenta accreta or placenta increta/percreta and antenatal diagnosis were similar to those of unsuspected women (Table 3).

## Discussion

### Main findings

This study showed that prevalence of PAS in the participating Italian regions was 0.84‰, with higher rates in Southern Italy.

Results highlighted the pivotal contribution of placenta previa or low-lying, prior CS and/ or other uterine surgery, and ART to the occurrence of PAS.

Half of the cases did not have antenatal suspicion, and this occurred also among women with relevant risk factors for PAS such as placenta previa and previous CS. Antenatal suspicion

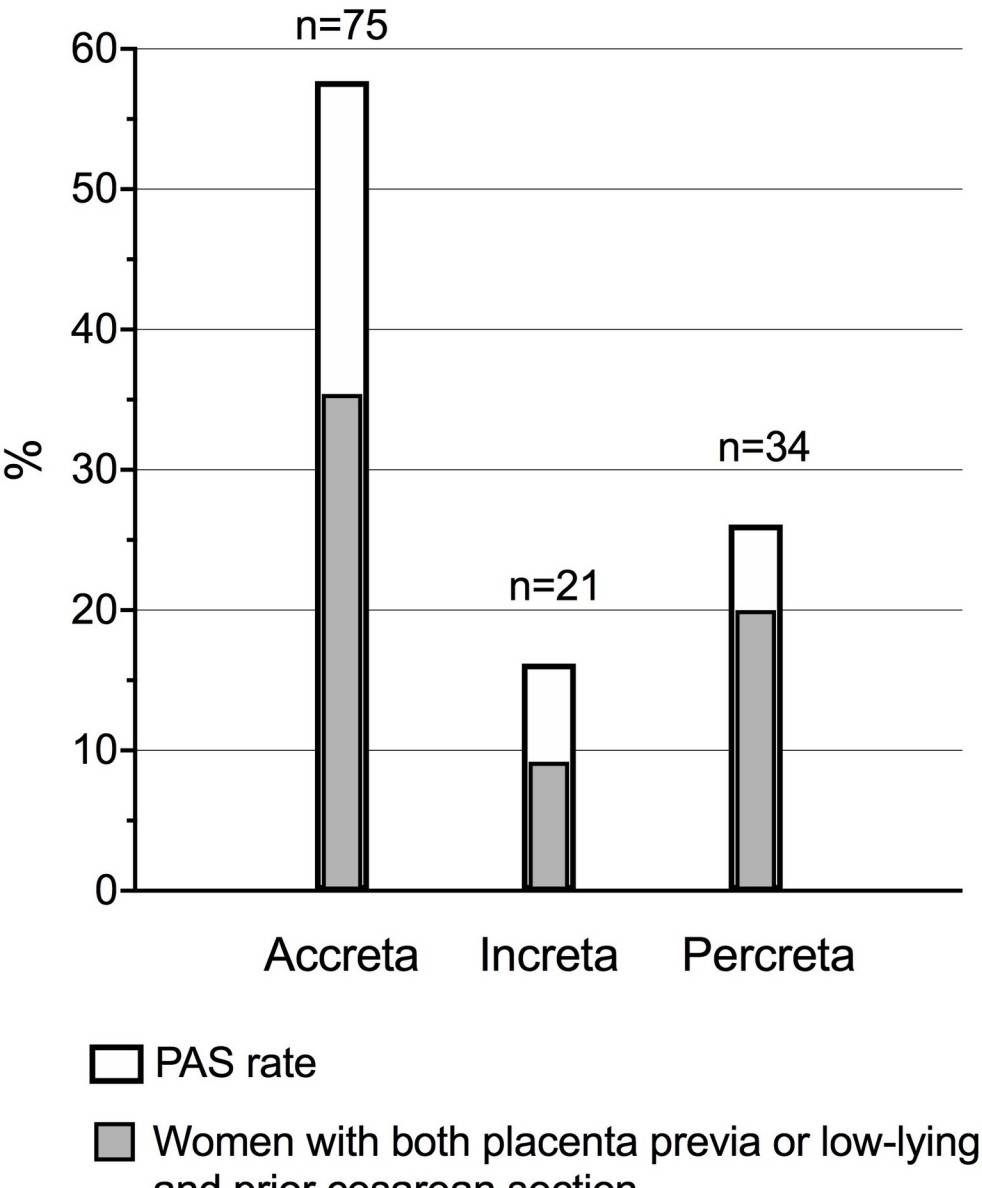

**Fig 2. Depth of placental invasion at histology.** Bar graphs show depth of placental invasion as defined by histopathology (white bars) and rates of concomitant placenta previa or low-lying and previous cesarean section (grey bars).

did not associate with improved outcomes in our cohort, except for a lower rate of RBC unit transfusion in suspected women. However, when assessed according to geographical area, adverse outcomes were less likely in patients in Southern regions notwithstanding higher rates of high-risk cases, such as those with both placenta previa and prior CS or placenta increta/ percreta.

### Strengths and limitations

The strengths of this study include the prospective and population-based design, the high participation rate of the maternity centers, and the opportunity to rely on the ItOSS surveillance

**Table 3. Postpartum bleeding and blood transfusion rates according to antenatal suspicion in patients with histologically diagnosed PAS.**

| Variables | Antenatal suspicion of PAS | | |
|---|---|---|---|
| **Placenta accreta (n = 75)** | **Yes** | **No** | *P*-value |
| | **N = 47 (62.7%)** | **N = 28 (37.3%)** | |
| | **N (%)** | **N (%)** | |
| PPH ≥2000 mL (4 missing) | 27 (58.7) | 15 (60.0) | 1.000 |
| RBC unit transfusion | 40 (85.1) | 26 (92.9) | 0.470 |
| • ≥4 RBC units | 21 (44.7) | 13 (46.4) | 1.000 |
| **Placenta increta or percreta (n = 55)** | **Yes** | **No** | |
| | **N = 38 (69.1%)** | **N = 17 (30.9%)** | |
| PPH ≥2000 mL (4 missing) | 8 (21.6) | 4 (28.6) | 0.715 |
| RBC unit transfusion | 28 (73.7) | 14 (82.4) | 0.733 |
| • ≥4 RBC units | 14 (36.8) | 5 (29.4) | 0.761 |

PPH, postpartum hemorrhage; RBC, Red Blood Cell.

system to validate the reported maternal death. Also, although subnational, results are unlikely to be significantly biased due the distribution of the participating regions in all the geographical areas of the country.

There are also limitations.

In order to fully capture all cases of PAS, we used a clinical case definition including also women with vaginal delivery. Although unlikely [34], this may have led to inclusion of cases of common entrapped placenta [35, 36] and thus, to overestimation of PAS prevalence. Of note, the present study was designed and implemented before FIGO guidelines on PAS definition were published [4].

Also, cases without histological confirmation of PAS were considered in the analyses. However, it is known that the absence of histological features indicative of PAS does not necessarily exclude such diagnosis, especially when high clinical suspicion is present [37].

In addition, information regarding blood loss at delivery was lacking in almost 16% of women and this may have led to biased result interpretation. Nevertheless, lack of missingness in first- and second line treatments of PPH has likely limited this possibility.

Another potential limitation is the lack of a PAS code in the ICD-9 Hospital Discharge database to ascertain completeness of notified cases. However, presence of a trained clinician in each hospital, the active monthly checks of ItOSS case reporting, and previous studies using ItOSS that suggested high rates of ascertainment [20], make this possibility unlikely.

Finally, the lack of individual data of deliveries in women without PAS prevented us from adjusting the estimated RRs.

## Interpretation

The PAS prevalence reported in this study (0.84‰) is higher than the one reported for the Nordic countries by the NOSS (0.34‰) [36]. This finding might be related to the exclusion of women with vaginal delivery from this work. However, a lower rate of PAS (0.46‰) was still identified when these women were included in a previous analysis [32], thus suggesting a more relevant role of prior CS rate (10% in Nordic countries vs 16.8% in Italy) in causing such a difference [20, 36]. Similarly, the higher rate of prior CS might explain the increased PAS prevalence in Italy compared to France (0.48‰, prior CS rate 11.4%) [38] and to the United Kingdom (0.17‰, prior CS rate 14.9%) [34]. In turn, the use of statistic record-linkage

procedures instead of active reporting may account for our lower prevalence estimates compared to a recent Australian-population based study (2.5‰, prior CS rate 14.4%) [16].

High rates of primary CS [39], alongside the policy defined by the axiom "once a cesarean always a cesarean" [40], has led to a considerable increase in women with ≥2 previous CS in the ItOSS cohort (96/384, 25%) compared to the Nordic countries (32/205, 15.6%). It is known that the incidence of placenta previa and of PAS rise with the number of prior CS [8–10]. According to this and in line with published data [36, 41], we identified a "dose-dependent" relation between prior CS and PAS, with an increase in the RR of PAS from 3.8 for one to 17.6 for ≥ two previous CS.

Antenatal suspicion of PAS has been associated to improved outcomes [18, 42–45]. This finding has also been confirmed by the UKOSS study [34]. Although our rate of antenatal suspicion (192/384, 50%) was similar to that reported in this work (66/134, 49.3%), we did not observe any difference in term of blood loss at delivery (Table 2), even when analysis was restricted to only those cases with histological confirmation of placenta increta or percreta (Table 3). Notwithstanding this, we noted a lower rate of RBC unit transfusion among suspected women, thus possibly suggesting a different preparedness to and, thus, management of severe PPH when substantial bleeding is expected at delivery and an adequate planning is put in place [34, 42, 46].

Of note, there were 25 (6.5%) women with both placenta previa or low-lying and prior CS, a combination of risk factors defining a clinical profile at high risk for adverse outcomes [38], who were not antenatally suspected. Knowledge of relevant risk factors for PAS is pivotal to guide a targeted prenatal ultrasound scan and increase the rate of antenatal diagnosis [42, 47–55]. However, PAS can also occur in the absence of any known risk factor, as we observed in 35 (9.1%) cases and as reported by the NOSS in 15 (7.3%) cases [36].

Almost half of our patients had a high-risk clinical profile [38]. Although such profile was substantially more common among women in the South compared to the North, we observed improved outcomes with decreased rates of PPH, blood transfusion, ICU admission, and severe maternal morbidity. Also, these women less frequently experienced intra- and post-hysterectomy complications, notwithstanding higher rates of placenta previa and previous CS, conditions known to make surgery more technically challenging [38, 44, 56, 57]. Since referral of expected cases has been suggested as a more important determinant of outcomes than the patient's clinical risk profile [45, 53, 58], it is plausible that this finding may be related to the higher rate of antenatal diagnosis observed among Southern women (61.7% vs 28.7%) with their subsequent referral to specialized centers, which occurred in 71.9% of the cases.

Altogether our results suggest that outcomes can be optimized even in women with a high-risk clinical profile when high rates of antenatal diagnosis are followed by referral to specialized centers with skilled multidisciplinary teams for PAS management.

Overall, almost 50% of women with PAS underwent hysterectomy in our study cohort, a rate similar to published data [34, 36]. Of note, a previous study from the same working group had reported PAS as the second leading cause (n = 191, 40.2%), after uterine atony (n = 214, 45.1%), of hysterectomy performed within 7 days of delivery for obstetric hemorrhage [20]. In the present work, all cases of PAS identified during the study period (n = 384) were included and assessed in terms of associated factors, management, and outcomes, providing novel Italian population-based data on the topic.

We observed a rate of severe maternal morbidity and peri-operative complications (13.3%) similar to that reported by the UKOSS (18/134, 13.4%) and the NOSS (21/205, 10.2%) [34, 36], with complications derived by severe hemorrhage being the most common.

Rate of maternal death in our study (2.6‰, national rate 0.09‰ [25]), was higher compared to the UKOSS, NOSS, and Australian cohorts, which did not report any fatal case [16, 34, 36],

but lower than the French cohort (4.1‰) [38]. Of note, the only death in our cohort occurred in an unsuspected woman without risk factors for PAS.

Also, we calculated a perinatal death rate of 20.8‰ (national rate 4.2‰ [59]) compared to 14.9, 9.8, 13.8, and 12 per 1000 of the UKOSS, NOSS, Australian, and French studies, respectively [16, 34, 36, 38]. Importantly, most of our cases were neonatal deaths occurring after very preterm delivery.

## Conclusions

A low CS rate in the population has been already proved as the most effective way to decrease CS-related adverse outcomes, including PAS [36, 60, 61]. Considering that Italy holds one of the highest rate of primary and elective repeat CS among European nations [28, 30], it is urgent to promote educational efforts to support Italian obstetricians in safely reducing primary CS and admitting women with prior CS to a trial of labor [29].

Management in specialized centers should be considered for all high-risk cases as pivotal determinant in improving outcomes [54, 55]. As recommended by the national guideline on PPH prevention and treatment [62], coordinated, multi-faceted efforts should be directed to increase antenatal suspicion of PAS by rising awareness of relevant risk factors with referral of patients at risk for targeted ultrasound assessment by expert sonographers [53, 63].

## Acknowledgments

We acknowledge the invaluable work of the Regional Obstetric Surveillance System Working Group, which include: Serena Donati, Alice Maraschini, Paola D'Aloja and Ilaria Lega (National Center for Disease Prevention and Health Promotion, Istituto Superiore di Sanità—Italian National Institute of Health), Vittorio Basevi (Center for Perinatal and Reproductive Health, Emilia-Romagna Region), Giuseppe Cali' (Department of Obstetrics and Gynecology, Civico Benfratelli Hospital, Palermo, Sicily), Gabriella Dardanoni (Health Department, Sicily Region), Valeria Dubini (Health Department, Tuscany Region), Camilla Lupi (Emilia-Romagna Region), Pasquale Martinelli (Department of Obstetrics and Gynecology, University of Naples Federico II), Luisa Mondo (Piedmont Region), Marcello Pezzella (Health Department, Campania Region), Monia Puglia (Health Department, Tuscany Region), Raffaella Rusciani (Piedmont Region), Daniela Spettoli (Emilia-Romagna Region), Fabio Voller (Health Department, Tuscany Region). The lead author of this group is Serena Donati; e-mail: serena.donati@iss.it.

This study would not have been possible without the enthusiasm and contribution of the ItOSS reporting clinicians who notified the cases and completed the data collection forms. We acknowledge Silvia Andreozzi for her valuable technical support.

## Author Contributions

**Conceptualization:** Alice Maraschini, Serena Donati.

**Data curation:** Sara Ornaghi, Alice Maraschini, Serena Donati.

**Formal analysis:** Sara Ornaghi, Alice Maraschini.

**Funding acquisition:** Serena Donati.

**Investigation:** Serena Donati.

**Methodology:** Alice Maraschini, Serena Donati.

**Project administration:** Alice Maraschini, Serena Donati.

**Resources:** Serena Donati.

**Supervision:** Serena Donati.

**Validation:** Sara Ornaghi, Alice Maraschini, Serena Donati.

**Visualization:** Sara Ornaghi.

**Writing – original draft:** Sara Ornaghi.

**Writing – review & editing:** Sara Ornaghi, Serena Donati.

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
