## [Decision Letter · Decision Letter 0]

27 Apr 2021

PONE-D-21-09866

Characteristics and outcomes of pregnant women with placenta accreta spectrum in Italy: a prospective population-based cohort study.

PLOS ONE

Dear Dr. Ornaghi,

Thank you for submitting your manuscript to PLOS ONE. After careful consideration, we feel that it has merit but does not fully meet PLOS ONE’s publication criteria as it currently stands. Therefore, we invite you to submit a revised version of the manuscript that addresses the points raised during the review process.

This is a well-written manuscript, although it requires some improvement. The authors state that all the relevant data are included with the manuscript, but unfortunately I could not find these at all. As it is a journal requirement this needs to be included in any revision. I would also like to see the Discussion include a (small) section on how the current manuscript enhances the findings published in reference [20]. Both reviewers have suggested other revisions that are required. Reviewer 1's point about the number of removals in the non-suspected group, and reviewer 2's point about how low the ratio of PAS suspected antenatally is, appear to be the most important to revise. However I agree with all their points so encourage dealing with them in any revision. I do not believe that these will be too taxing to complete.

We look forward to receiving your revised manuscript.

Kind regards,

Clive J Petry, PhD

Academic Editor

PLOS ONE

Journal Requirements:

3. In the ethics statement in the manuscript and in the online submission form, please provide additional information about the patient records used in your retrospective study, including: a) whether all data were fully anonymized before you accessed them; b) the date range (month and year) during which patients' medical records were accessed; c) the date range (month and year) during which patients whose medical records were selected for this study sought treatment. If the ethics committee waived the need for informed consent, or patients provided informed written consent to have data from their medical records used in research, please include this information.

4. One of the noted authors is a group or consortium [Regional Obstetric Surveillance System Working Group]. In addition to naming the author group, please list the individual authors and affiliations within this group in the acknowledgments section of your manuscript. Please also indicate clearly a lead author for this group along with a contact email address.

Reviewers' comments:

Reviewer's Responses to Questions

**Comments to the Author**

1. Is the manuscript technically sound, and do the data support the conclusions?

Reviewer #1: Yes

Reviewer #2: Yes

2. Has the statistical analysis been performed appropriately and rigorously? 

Reviewer #1: Yes

Reviewer #2: Yes

3. Have the authors made all data underlying the findings in their manuscript fully available?

Reviewer #1: Yes

Reviewer #2: No

4. Is the manuscript presented in an intelligible fashion and written in standard English?

Reviewer #1: Yes

Reviewer #2: No

5. Review Comments to the Author

Reviewer #1: Congratulation with a well written manuscript.

I have several concerns:

1. high number of missing values: on 61 women the blood loss is not known. This should be reported in the strength & limitation section

2. In table 1 the part on placenta previa: NOT having a placenta previa gives a RR of 227??? Please revise. Is this number correct? Or does it belong to YES having a placenta previa?

3. The number of manual removals in the non-suspected group is much higher than in the group with PAS suspicion: are we looking at another degree of PAS?

4. Alternative ways to handle PAS like focal resection or leaving the placenta in situ (in case of percreta) are not mentioned. Are they not performed? Please comment on this in your manuscript

Reviewer #2: Thanks to the authors for this large-scale study.

Shortly;

In this study, the authors reported that placenta acreta spectrum (PAS) is a life-threatening condition that can lead to severe bleeding, the prevalence increases as the caesaren rates increase, the caesarean rates are high in Italy and the number of quality studies on PAS is low. They were stated that the study was part of a large population-based study on near miss events caused by bleeding, conducted by the Italian obstetric survey system. The aim of the article was to determine the incidence of PAS, to investigate the management of associated factors and perinatal complications, and the results were compared by regions, since there were regions that could not access health services in a balanced way. The authors reported that the study was conducted with about 212 units in a total of six regions from three north and three in the south with an annual birth rate of over 25 000 each. Women with a diagnosis of PAS between the ages of 15-50 over the 22nd gestational week between September 2014 and 2016 were included in the study. This population covers 50% of all births in Italy. As a result of the study, they found the incidence of PAS ‰ 0.84, and reported that although the risk of cases such as placenta previa, low lying placenta, PAS was higher in the southern regions, the possibility of adverse outcomes was lower.

Here is my review and recommendations;

1. Material and methods section; line 196-202

Were the patients in Items 1, 2, 3 also included in the PAS group, or were only the patients specified in Item 4 considered PAS. If so;

Line 196-197 de ‘’severe postpartum hemorrhage, defined as “hemorrhage within 7 days from delivery requiring ≥4 units of whole blood or packed red blood cells ‘’ How was PAS diagnosed in these patients?

Line 197-198 ‘’hemorrhage due to complete or incomplete uterine rupture” How did you rule out non-PAS causes of uterine rupture?

Line 198-199 da (3)“peripartum hysterectomy within 7 days from delivery” How was the diagnosis of PAS made in those without histological examination?

2. Line 236 -238 ‘’Figure 1: Bar graphs show regional and overall prevalence distribution of PAS (white bars), as well as of concomitant placenta previa or low-lying and previous cesarean section among women with PAS’’

Descriptions in the text and explanations below the figure are incompatible

When Figure 1 is examined, it is not fully understood.

White bars: If '' Placenta previa or low-lying and prior CS ''

Gray bars: Cases without PAS risk? Are there vaginal births in this group?

Gray and white bars: All cases of PAS?

Can you please explain and correct in text.

Can you please rearrange Figure 1 in an understandable way?

3. Line 260 ‘’Delivery occurred in facilities in Southern Italy in 65% of the cases’’ is written, but in table 1 this ratio is given as 35.4. Which one is right? Please correct.

4. Line 278 , ‘’PAS was antenatally suspected in 50% of the cases’’

Isn't this ratio too low? Today, evaluation of placenta during fetal anatomical scanning and confirmation at 26th gestational weeks are recommended in various guideline.

Line 419-421 ‘’Knowledge of relevant risk factors for PAS is pivotal to guide a targeted prenatal ultrasound scan and increase the rate of antenatal diagnosis’’

The PAS predicton might be possible in antenatal period with USG and doppler findins such as the loss of the clear zone, presence of placental lacunae, and interruption of the bladder-uterus border according to grey-scale ultrasonography, and increased vascularity in this area based on Doppler USG. According to this, I think 50% is less for the PAS prediction. Isn't antenatal USG performed for every patient in Italy, is it only performed for those with risk factors?

Readers will want to know about routine antenatal care in Italy. Please provide at least one or two paragraphs of information about this.

5. Line 354 Table 3. The title is not clear, does not describe the table

Please change as ‘’Postpartum bleeding and blood transfusion rates according to antenatal suspicion in patients with histologically diagnosed PAS’’

6. There are lots of expression disorders and grammatical errors in the article, so English editing and proofreading should be done.

6. PLOS authors have the option to publish the peer review history of their article (what does this mean?). If published, this will include your full peer review and any attached files.

Reviewer #1: No

Reviewer #2: No

---

## [Author Response · Author response to Decision Letter 0]

9 May 2021

Manuscript requests from the Academic Editor Dr. Clive J Petry, PhD.

We have included a revised statement regarding data availability.

Precisely, data cannot be shared publicly because of ethical standards and legal requirements. Data are available from the Ethics Committee of the Istituto Superiore di Sanità - Italian National Institute of Health (contact via email: segreteria.comitatoetico@iss.it) for researchers who meet the criteria for access to confidential data.

We have now included a brief paragraph in the Discussion section detailing how the findings reported in the current manuscript differentiate from and enhance those published by the same working group regarding peripartum hysterectomy due to obstetric hemorrhage within seven days from delivery in Italy (Maraschini et al., Acta obstetricia et gynecologica Scandinavica. 2020).

Journal requirements

1. We ensure that our manuscript meets PLOS ONE’s style requirements.

2. The reference list has been reviewed and we ensure it is complete and correct. None of the cited papers has been retracted.

3. Women included in the study sought treatment in the participating centers between September 1st, 2014 and August 31st, 2016. Data were fully anonymized before being accessed and analyzed. Since data were analyzed anonymously, need for informed consent was waived by the Ethics Committee of the Istituto Superiore di Sanità - Italian National Institute of Health. 

4. All the individual authors of the Regional Obstetric Surveillance System Working Group have now been added, alongside their affiliations, to the Acknowledgments section of our manuscript. The lead author for this group is the senior author of the manuscript, Dr. Serena Donati. We have now specified this information and added her contact email address.

Dear Referees:

Thank you for the strong positive comments on the first submission of our manuscript. We also appreciate the helpful suggestions as to how we might further improve the paper. Below we respond to each suggestion with details about how we amended the manuscript.

Response to referees.

Response to referee 1. 

We appreciate the comment from referee 1 congratulating us on a well written manuscript.

The referee pointed out the high number (61/384, 15.9%) of missing values for the variable ‘blood loss at delivery’ and how this should be highlighted in the Strengths and Limitations section of our manuscript. 

Considering the potential for bias due to >10% missingness, we have now included this as a limitation of our research work. However, we also highlighted that the lack of missingness in data regarding first- and second line treatments of PPH has possibly limited the chance of a biased result interpretation. 

Referee 1 asked about the role of placenta previa or low-lying as risk factor for PAS since Table 1 shows a relative risk of 227 associated with not having placenta previa or low-lying. 

We apologize for the mistake; the relative risk of 227 is associated with having the condition. This has now been corrected in Table 1. 

The referee observed that the number of manual removals in the non-suspected group is much higher than in the group with PAS suspicion and this could be the clinical manifestation of another degree of PAS.

We agree with the referee’s observation. Yet, we would like to point out that unsuspected women were also more likely to receive second-line uterotonic drugs, whereas women with antenatal suspicion were more frequently managed with vascular embolization and hysterectomy (Table 2). These data suggest that in the clinical context of a retained placenta, with or without hemorrhage, clinicians might be more prone to manage the condition as subsequent to uterine atony (the most frequent cause) if no antenatal suspicion of PAS is present. Of note, thirty out of 32 unsuspected women with manual removal of the placenta in our cohort delivered vaginally. In addition, when assessing histologically confirmed PAS cases, we did not identify any difference regarding the grade of invasion between women with and without antenatal suspicion, although we are aware that lack of histological report in some of the cases might have affected these findings. 

Referee 1 mentioned that alternative ways to hysterectomy for PAS management, including focal resection and leaving the placenta in situ (for cases of placenta percreta), exist but these were not reported in our manuscript. 

In our cohort there were four women managed conservatively: three had a partial placenta accreta diagnosed after delivery and only the abnormally adherent cotyledon was left in situ whereas the remaining one had an antenatal diagnosis of complete placenta previa with signs of percretion and no attempt of placenta removal was performed at the time of cesarean section. There were no women in whom the focal resection technique was applied. 

We have now included these data in the Results section.

Response to referee 2. 

We thank the referee 2 for highlighting that this is a large-scale, population-based study, which can provide useful information to guide national initiatives.

Referee 2 asked whether patients with (1) severe postpartum hemorrhage, defined as “hemorrhage within 7 days from delivery requiring ≥4 units of whole blood or packed red blood cells”; (2) “hemorrhage due to complete or incomplete uterine rupture”; (3) “peripartum hysterectomy within 7 days from delivery”; and (4) PAS, clinically defined as “difficult or incomplete manual removal of the placenta following vaginal delivery and the need of blood transfusion within 48 hours” or “difficult removal of the placenta during cesarean delivery and clinical evidence of an abnormally invasive placenta”, were all included in the present study. 

If not, the referee was wondering how PAS was diagnosed in patients with (1), (2), and (3).

PAS was diagnosed clinically, as defined in (4). All women reported as having PAS by the reference clinician in the participating centers were included in the present study, independent of the associated outcomes, such as severe postpartum hemorrhage (1), uterine rupture (2), and peripartum hysterectomy within 7 days from delivery (3), leading to inclusion in the wider ItOSS research project on severe maternal morbidity due to obstetric hemorrhage.

This has been now clarified in the text. 

The referee suggested we provide a better description of Figure 1 in the text and we rearrange it in a more understandable way. 

We thank the Referee for this comment. We have now included a brief description of Figure 1 in the Results section and amended both the Figure and its caption to improve clarity.

Referee 2 pointed out that Table 1 reports a 35.4% rate of delivery in Southern Italy. 

This was a mistake: rate of delivery in Southern Italy was 64.6% whereas rate of delivery in Northern Italy was 35.4%. This has now been corrected in Table 1. 

The referee was wondering whether the reported rate of antenatal suspicion (50%) might be too low, considering the current ultrasound techniques that allow an in-depth investigation of the placenta during gestation. 

We agree with the referee that a detailed ultrasound scan, particularly in high-risk women, is pivotal in antenatally suspecting PAS through identification of suggestive signs, such as the loss of the clear zone, the presence of placental lacunae, and the interruption of the bladder-uterus border at grey-scale investigation, and the increased vascularity in the area at Doppler assessment. We also recognize that these ultrasound signs are included in several guidelines on the topic. 

However, we kindly disagree with the referee suggesting that a 50% rate of antenatal diagnosis is too low. This rate is similar to that reported by the UKOSS study (66/134, 49.3%) (Fitzpatrick et al., 2014) and the French study (106/242, 43.8%), (Kayem et al., 2021) and substantially higher than that of the NOSS study (60/205, 29.3%) (Thurn et al., 2016), which all had a time period for women’s inclusion in the project similar to ours. 

It is likely that currently ongoing research works on PAS, including women in more recent years, might show higher rates of antenatal suspicion, given the increased prevalence and thus awareness of this condition and the institution of dedicated PAS diagnostic services for high-risk women in tertiary care centers, as recently reported by Bhide and colleagues. (Coutinho et al., 2021).

Referee 2 suggested to modify the title of Table 3 to improve clarity. 

We appreciate the referee’s suggestion and we have now changed the title as per the referee’s indications. 

The referee pointed out the presence of expression disorders and grammatical errors in the manuscript.

The manuscript has now been proofread by a native English speaker, thus hopefully improving its readability. 

References (almost all of these citations are also included in the manuscript)

Coutinho CM, Giorgione V, Noel L, Liu B, Chandraharan E, Pryce J, Frick AP, Thilaganathan B, Bhide A (2021) Effectiveness of contingent screening for placenta accreta spectrum disorders based on persistent low-lying placenta and previous uterine surgery. Ultrasound Obstet Gynecol 57:91-96.

Fitzpatrick KE, Sellers S, Spark P, Kurinczuk JJ, Brocklehurst P, Knight M (2014) The management and outcomes of placenta accreta, increta, and percreta in the UK: a population-based descriptive study. Bjog 121:62-70; discussion 70-61.

Kayem G et al. (2021) Clinical profiles of placenta accreta spectrum: the PACCRETA population-based study. Bjog.

Thurn L, Lindqvist PG, Jakobsson M, Colmorn LB, Klungsoyr K, Bjarnadóttir RI, Tapper AM, Børdahl PE, Gottvall K, Petersen KB, Krebs L, Gissler M, Langhoff-Roos J, Källen K (2016) Abnormally invasive placenta-prevalence, risk factors and antenatal suspicion: results from a large population-based pregnancy cohort study in the Nordic countries. Bjog 123:1348-1355.

---

## [Editor Report · Decision Letter 1]

20 May 2021

Characteristics and outcomes of pregnant women with placenta accreta spectrum in Italy: a prospective population-based cohort study.

PONE-D-21-09866R1

Dear Dr. Ornaghi,

We’re pleased to inform you that your manuscript has been judged scientifically suitable for publication and will be formally accepted for publication once it meets all outstanding technical requirements.

Kind regards,

Clive J Petry, PhD

Academic Editor

PLOS ONE
---

## [Editor Report · Acceptance letter]

25 May 2021

PONE-D-21-09866R1 

Characteristics and outcomes of pregnant women with placenta accreta spectrum in Italy: a prospective population-based cohort study. 

Dear Dr. Ornaghi:

I'm pleased to inform you that your manuscript has been deemed suitable for publication in PLOS ONE. Congratulations! Your manuscript is now with our production department. 

Kind regards, 

on behalf of

Dr. Clive J Petry 

Academic Editor

PLOS ONE